# Improved Feature Distillation via Projector Ensemble

**Yudong Chen**[1]   **Sen Wang**[1]*   **Jiajun Liu**[2]*   **Xuwei Xu**[1]   **Frank de Hoog**[2]   **Zi Huang**[1]

[1]The University of Queensland    [2]CSIRO Data61

{yudong.chen,sen.wang,xuwei.xu}@uq.edu.au
{jiajun.liu,frank.dehoog}@csiro.au
huang@itee.uq.edu.au

## Abstract

In knowledge distillation, previous feature distillation methods mainly focus on the design of loss functions and the selection of the distilled layers, while the effect of the feature projector between the student and the teacher remains under-explored. In this paper, we first discuss a plausible mechanism of the projector with empirical evidence and then propose a new feature distillation method based on a projector ensemble for further performance improvement. We observe that the student network benefits from a projector even if the feature dimensions of the student and the teacher are the same. Training a student backbone without a projector can be considered as a multi-task learning process, namely achieving discriminative feature extraction for classification and feature matching between the student and the teacher for distillation at the same time. We hypothesize and empirically verify that without a projector, the student network tends to overfit the teacher's feature distributions despite having different architecture and weights initialization. This leads to degradation on the quality of the student's deep features that are eventually used in classification. Adding a projector, on the other hand, disentangles the two learning tasks and helps the student network to focus better on the main feature extraction task while still being able to utilize teacher features as a guidance through the projector. Motivated by the positive effect of the projector in feature distillation, we propose an ensemble of projectors to further improve the quality of student features. Experimental results on different datasets with a series of teacher-student pairs illustrate the effectiveness of the proposed method. Code is available at `https://github.com/chenyd7/PEFD`.

## 1   Introduction

The last decade has witnessed the rapid development of Convolutional Neural Networks (CNNs) [19, 29, 9, 20]. The resulting increases in performance however, have come with substantial increases in network size and this largely limits the applications of CNNs on edge devices [13]. To alleviate this problem, knowledge distillation has been proposed for network compression. The key idea of distillation is to use the knowledge obtained by the large network (teacher) to guide the optimization of the lightweight network (student) [12, 24, 31].

Existing methods can be roughly categorized into logit-based, feature-based and similarity-based distillation [7]. Recent research shows that feature-based methods generally distill a better student network compared to the other two groups [30, 4]. We conjecture that the process of mimicking the teacher's features provides a clearer optimization direction for the training of the student network. Despite the promising performance of feature distillation, it is still challenging to narrow the gap between the student and teacher's feature spaces. To improve the feature learning ability of the

---

*Corresponding Author

36th Conference on Neural Information Processing Systems (NeurIPS 2022).

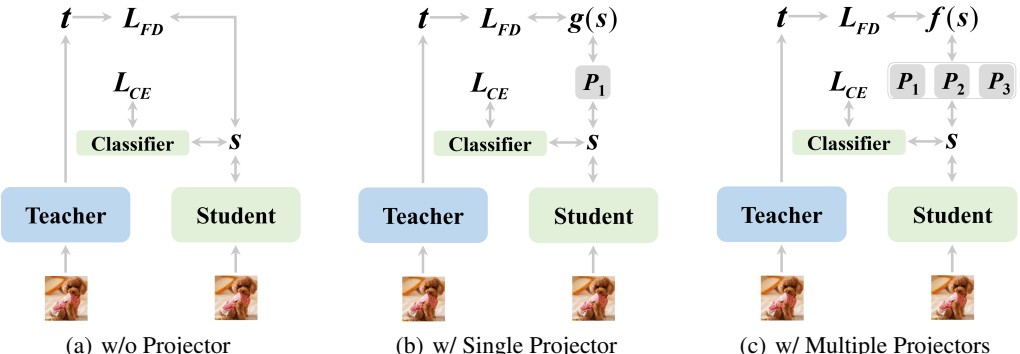

(a) w/o Projector      (b) w/ Single Projector      (c) w/ Multiple Projectors

Figure 1: Illustration of (a) feature distillation without a projector when the feature dimensions of the student and the teacher are the same, (b) the general feature-based distillation with a single projector [4, 33] and (c) the proposed method with multiple projectors, where $\mathcal{L}_{CE}$ and $\mathcal{L}_{FD}$ are the cross-entropy loss and the feature distillation loss, respectively.

student, various feature distillation methods have been developed by designing more powerful objective functions [30, 36, 33, 4] and determining more effective links between the layers of the student and the teacher [2, 15, 1].

We have found that the feature projection process from the student to the teacher's feature space plays a key part in feature distillation and can be redesigned to improve the performance. Since the feature dimensions of student networks are not always the same with that of teacher networks, a projector is often required to map features into a common space for matching. As shown in Table 1, imposing a projector on the student network can improve the distillation performance even if the feature dimensions of the student and the teacher are the same. We hypothesize that adding a projector for distillation helps to mitigate the overfitting problem when minimizing the feature discrepancy between the student and the teacher. As shown in Figure 1(a), distillation without a projector can be regarded as a multi-task learning process, including feature learning for classification and feature matching for distillation. In this case, the student network may overfit the teacher's feature distributions and the generated features are less distinguishable for classification. Our empirical results in Section 3 support this hypothesis to some extent. Besides, inspired by the effectiveness of adding a projector for feature distillation, we propose an ensemble of projectors for further improvement. Our intuition is that projectors with different initialization generate diverse transformed features. Therefore, it is helpful to improve the generalization of the student by using multiple projectors according to the theory behind ensemble learning [37, 32]. Figure 1 shows the comparisons of existing distillation methods and our method.

Our contributions are three-fold:

- We investigate the phenomenon that the student benefits from a projector during feature distillation even when the student and the teacher have identical feature dimensionalities.
- Technically, we propose an ensemble of feature projectors to improve the performance of feature distillation. The proposed method is extremely simple and easy to implement.
- Experimentally, we conduct comprehensive comparisons between different methods on benchmark datasets with a wide variety of teacher-student pairs. It is shown that the proposed method consistently outperforms state-of-the-art feature distillation methods.

## 2 Related Work

Since this paper mainly focuses on the design of the projector, we divided the existing methods into two categories in term of the usage of the projector as follows:

**Projector-free methods.** As the most representative distillation method, Knowledge Distillation (KD) [12] proposes to utilize the logits generated by the pre-trained teacher to be the additional targets of the student. The intuition of KD is that the generated logits are able to provide more

useful information than the general binary labels for optimization. Motivated by the success of KD, various logit-based methods have been proposed for further improvement. For example, Deep Mutual Learning (DML) [35] proposes to replace the pre-trained teacher with an ensemble of students so that the distillation mechanism does not need to train a large network in advance. Teacher Assistant Knowledge Distillation (TAKD) [21] observes that a better teacher may distill a worse student due to the large performance gap between them. Therefore, a teacher assistant network is introduced to alleviate this problem. Another technical route of projector-free methods is the similarity-based distillation. Unlike the logit-based methods that aim to exploit the category information hidden in the predictions of the teacher, similarity-based methods try to explore the latent relationships between samples in feature space. For example, Similarity-Preserving (SP) [31] distillation first constructs the similarity matrices of the student and the teacher by computing the inner products between features and then minimises the discrepancy between the obtained similarity matrices. Similarly, Correlation Congruence (CC) [23] forms the similarity matrices with a kernel function. Although the logit-based and similarity-based methods do not require an extra projector during training, they are generally less effective than the feature-based methods as shown in the recent research [4, 33].

**Projector-dependent methods.** Feature distillation methods aim to make student and teacher features as similar as possible. Therefore, a projector is essential to map features into a common space. The first feature distillation method FitNets [24] minimizes the L2 distance between student and teacher feature maps produced by the intermediate layer of networks. Furthermore, Contrastive Representation Distillation (CRD) [30], Softmax Regression Representation Learning (SRRL) [33] and Comprehensive Interventional Distillation (CID) [4] show that the last feature representations of networks are more suitable for distillation. One potential reason is that the last feature representations are closer to the classifier and will directly affect the classification performance [33]. The aforementioned feature distillation methods mainly focus on the design of loss functions such as introducing contrastive learning [30] and imposing causal intervention [4]. A simple 1x1 convolutional kernel or a linear projection is adopted to transform features in these methods. We note that the effect of projectors is largely ignored. Previous works such as Factor Transfer (FT) [16] and Overhaul of Feature Distillation (OFD) [11] try to improve the architecture of projectors by introducing the auto-encoder and modifying the activation function. However, their performance is not competitive when compared to the state-of-the-art methods [33, 4]. Instead, this paper proposes a simple distillation framework by combining the ideas of distilling the last features and projector ensemble.

## 3  Improved Feature Distillation

We first define the notations used in the following sections. In line with observations in recent research [30, 4], we apply the feature distillation loss to the layer before the classifier. $S = \{s_1, s_2, ..., s_i, ..., s_b\} \in \mathbb{R}^{d \times b}$ denotes the last student features, where $d$ and $b$ are the feature dimension and the batch size, respectively. The corresponding teacher features are represented by $T = \{t_1, t_2, ..., t_i, ..., t_b\} \in \mathbb{R}^{m \times b}$, where $m$ is the feature dimension. To match the dimensions of $S$ and $T$, a projector $g(\cdot)$ is required to transform the student or teacher features. We experimentally find that imposing the projector on the teacher is less effective since the original and more informative feature distribution from the teacher would be disrupted. Therefore, in the proposed distillation framework, a projector will be added to the student as $g(s_i) = \sigma(W s_i)$ during training and be removed after training, where $\sigma(\cdot)$ is the ReLU function and $W \in \mathbb{R}^{m \times d}$ is a weighting matrix.

### 3.1  Feature Distillation as Multi-task Learning

In recent work, SRRL and CID combine the feature-based loss with the logit-based loss to improve the performance. Since distillation methods are sensitive to hyper-parameters and changes of teacher-student combinations, the additional objectives will increase the training cost for coefficients adjustment. To alleviate this problem, we simply use the following Direction Alignment (DA) loss [17, 3, 8] for feature distillation:

$$\mathcal{L}_{DA} = \frac{1}{2b} \sum_{i=1}^{b} ||\frac{g(s_i)}{||g(s_i)||_2} - \frac{t_i}{||t_i||_2}||_2^2 = 1 - \frac{1}{b} \sum_{i=1}^{b} \frac{\langle g(s_i), t_i \rangle}{||g(s_i)||_2 ||t_i||_2}, \quad (1)$$

where $||\cdot||_2$ indicates the L2-norm and $\langle \cdot, \cdot \rangle$ represents the inner product of two vectors. By convention [12, 30, 33], the distillation loss is coupled with the cross-entropy loss to train a student. As mentioned

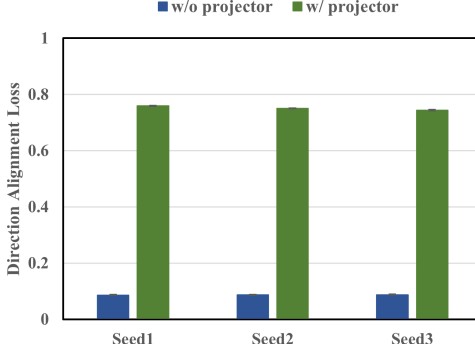
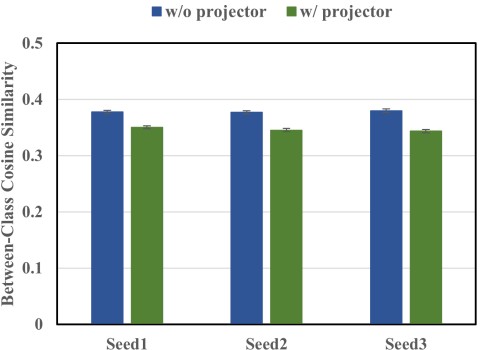

Figure 2: The left figure displays the direction alignment loss between teacher and student features with and without a projector. The right figure displays the average between-class cosine similarities in students' feature spaces. These results are obtained on CIFAR-100 with a teacher-student pair ResNet32x4-ResNet8x4. The feature dimensions of ResNet32x4 and ResNet8x4 are the same, which means $m = d$.

in the Introduction, adding a projector helps to improve the distillation performance even if the feature dimensions of the student and the teacher are the same. We hypothesize that the training process of the student network without a projector can be viewed as a multi-task learning (i.e. distillation and classification tasks) in the same feature space. As such, student features tend to overfit teacher features and become less discriminative for classification. We use two measurements to empirically verify this hypothesis. The first one is to measure the discrepancy between student and teacher features as follows:

$$\mathcal{M}_{DA} = 1 - \frac{1}{b} \sum_{i=1}^{b} \frac{\langle s_i, t_i \rangle}{||s_i||_2 ||t_i||_2}. \tag{2}$$

We display the $\mathcal{M}_{DA}$ results of students with and without a projector in Figure 2. Apparently, since student features will directly interact with teacher features, the $\mathcal{M}_{DA}$ results of the student without a projector are significantly lower than that of the student with a projector in different seeds. However, we found that the distilled student features without a projector is less discriminative by investigating the between-class cosine similarity in the student's feature space. The between-class cosine similarity is computed as follows:

$$\mathcal{M}_{BC} = \frac{1}{b} \sum_{i=1}^{b} \sum_{j=1}^{c_i} \frac{\langle s_i, s_j \rangle}{c_i ||s_i||_2 ||s_j||_2}, \tag{3}$$

where $s_j$ is the $j$-th sample belonging to a class different from that of $s_i$ and $c_i$ is the total number of $s_j$ corresponding to $s_i$. The results of $\mathcal{M}_{BC}$ are in Figure 2. It is shown that the student network with a projector generate more discriminative features compared to the student network without a projector. The results of $\mathcal{M}_{DA}$ and $\mathcal{M}_{BC}$ suggest that the student tends to overfit the teacher's feature space without a projector. Since the tasks of classification and distillation are performed in the same feature space, the generated features are less distinguishable for classification. By adding a projector to disentangle the two tasks, we can improve the classification performance of the student.

## 3.2 Improved Feature Distillation with Projector Ensemble

The above analysis suggests that the projector can boost the distillation performance of the student. Motivated by this, we propose an ensemble of projectors for further improvement. There are two motivations of using multiple projectors. Firstly, projectors with different initialization provide different transformed features, which is beneficial to the generalizability of the student [37, 32]. Secondly, since we use the ReLU function to enable the projector to perform nonlinear feature extraction, the projected student features may contain zeros. On the other hand, teacher features are less likely to be zeros due to the commonly used average pooling operation in CNNs. That is to say, the feature distribution gap between the teacher and the student is large with a single projector. Therefore, using ensemble learning is a natural way to achieve a good trade-off between training

**Algorithm 1** Improved Feature Distillation via Projector Ensemble.
___
**Input:** The pre-trained teacher, the structure of the student, training data $X$ and labels.
**Parameter:** Total iterations $N$, $\alpha$ and the number of projectors $q$.
**Initialization:** Initialize different projectors and the student.
**Training:**
1: **for** $i = 1 \rightarrow N$ **do**
2:  Sample a mini-batch data from $X$.
3:  Generate $S$, $T$ and the student's prediction by forward propagation.
4:  Update projectors and the student network by minimizing objective (5).
5: **end for**
**Output**: The distilled student.
___

error and generalizability. By introducing multiple projectors, the Modified Direction Alignment (MDA) loss is as follows:

$$\mathcal{L}_{MDA} = 1 - \frac{1}{b} \sum_{i=1}^{b} \frac{\langle f(s_i), t_i \rangle}{||f(s_i)||_2 ||t_i||_2}, \tag{4}$$

where $f(s_i) = \frac{1}{q} \sum_{k=1}^{q} g_k(s_i)$, $q$ is the number of projectors and $g_k(\cdot)$ indicates the $k$-th projector. By combining the distillation loss (4) and the classification loss together, we obtain the following objective function to train the student:

$$\mathcal{L}_{total} = \mathcal{L}_{CE} + \alpha \mathcal{L}_{MDA}, \tag{5}$$

where $\mathcal{L}_{CE}$ is the cross-entropy loss and $\alpha$ is a hyper-parameter. The details of our method are shown in Algorithm 1.

## 4 Experiments

We conduct comprehensive experiments to evaluate the performance of different methods and the effectiveness of the proposed projector ensemble-based feature distillation, on image classification task. Implementation details are as follows:

**Baselines.** We select representative distillation methods in various categories for comparisons, including logit-based method KD [12], similarity-based methods CC [23], SP [31] and Relational Knowledge Distillation (RKD) [22], feature-based methods FitNets [24], FT [16], CRD [30], SRRL [33] and CID [4]. The logit-based and similarity-based methods are projector-free and the feature-based methods require additional projectors. FitNets and SRRL use convolutional kernels to transform the student features. FT adopts an auto-encoder to extract the latent feature representations of the student and the teacher. CRD maps student and teacher features into a low-dimensional space while CID maps student features into the teacher space with linear projections. For simplicity, the proposed method constructs the projector by combining a linear projection and the ReLU function.

**Datasets.** Two benchmark datasets are used for evaluation in our experiments. ImageNet [25] contains approximately 1.28 million training images and 50,000 validation images from 1,000 classes. The validation images are used for testing. Each image is resized to 224x224. CIFAR-100 [18] dataset includes 50,000 training images and 10,000 testing images from 100 classes. Each image is resized to 32x32. On ImageNet and CIFAR-100, we adopt the commonly used random crop and horizontal flip techniques for data augmentation.

**Teacher-student pairs.** To validate the generalizability of different distillation methods, we select a group of popular network architectures to form different teacher-student pairs. The teacher networks include ResNet34 [9], DenseNet201 [14], WRN-40-2 [34], VGG13 [27], ResNet32x4 [9] and ResNet50 [9]. The student networks comprise of ResNet18 [9], MobileNet [13], WRN-16-2 [34], VGG8 [27], ResNet8x4 [9] and MobileNetV2 [26]. By combining different teacher and student networks, we can perform distillation between similar architectures (e.g., ResNet34-ResNet18) and different architectures (e.g., DenseNet201-ResNet18).

**Training.** Following the settings of previous methods[2], the batch size, epochs, learning rate decay rate and weight decay rate are 256/64, 100/240, 0.1/0.1, and 0.0001/0.0005, respectively on

___
[2]https://github.com/HobbitLong/RepDistiller

Table 1: Top-1 classification accuracy (%) on CIFAR-100 using horizontal ensembles with different teacher-student pairs. 1-Proj, 2-Proj, 3-Proj and 4-Proj indicate the number of single-layer projectors in the ensemble.

| Pair | Student | w/o Proj | 1-Proj | 2-Proj | 3-Proj | 4-Proj | Teacher |
|---|---|---|---|---|---|---|---|
| VGG13-VGG8 | 70.74 | 73.76 | 73.84 | 74.21 | **74.35** | 74.18 | 74.64 |
| ResNet32x4-ResNet8x4 | 72.93 | 73.66 | 75.14 | 75.66 | **76.08** | 75.93 | 79.42 |

Table 2: Top-1 classification accuracy (%) on CIFAR-100 using deep projectors with different teacher-student pairs. 2-MLP, 3-MLP and 4-MLP indicate the depth of the projectors with different number of layers.

| Pair | Student | w/o Proj | 1-Proj | 2-MLP | 3-MLP | 4-MLP | Teacher |
|---|---|---|---|---|---|---|---|
| VGG13-VGG8 | 70.74 | 73.76 | **73.84** | 73.31 | 73.02 | 72.73 | 74.64 |
| ResNet32x4-ResNet8x4 | 72.93 | 73.66 | **75.14** | 75.12 | 74.56 | 74.30 | 79.42 |

**ImageNet/CIFAR-100.** The initial learning rate is 0.1 on ImageNet, and 0.01 for MobileNetV2, 0.05 for the other students on CIFAR-100. Besides, the learning rate drops at every 30 epochs on ImageNet and drops at 150, 180, 210 epochs on CIFAR-100. The optimizer is Stochastic Gradient Descent (SGD) with momentum 0.9. All the experiments are performed on an NVIDIA V100 GPU.

**Hyper-parameters.** By following the conventions in CRD [30], we use the same settings for the hyper-parameters of KD, CC, SP, RKD, FitNets, FT and CRD. For SRRL and CID, the settings of hyper-parameters are provided by the corresponding authors. For the proposed method, we set $\alpha = 25$ and $q = 3$ by tuning with teacher-student pair ResNet34-ResNet18 on ImageNet. For a fair comparison, the hyper-parameters of different methods are fixed in all experiments.

## 4.1 Ablation Studies

This section studies the effectiveness of the proposed projector ensemble method, and how different ensemble strategies affect the performance. In this experiment, two different network architectures, i.e. VGG-style and ResNet-style networks are used for illustration in Tables 1, 2 and 3.

**Horizontal ensemble of projectors.** Table 1 shows the top-1 classification accuracy of the proposed projector ensemble with different number of projectors. It verifies that imposing a projector improves the distillation performance when the feature dimensions of the student and the teacher are the same. A potential reason is that the projector helps to disentangle the two learning tasks (i.e. distillation and classification) and improves the quality of student features. In addition, by integrating multiple projectors, the proposed method further increases the classification accuracy by a clear margin with various numbers of projectors.

**Deep cascade of projectors.** Another common way to modify the architecture is to increase the depth of the projector. Table 2 demonstrates the changes of distillation performance by gradually stacking non-linear projections. In this table, 2-MLP, 3-MLP and 4-MLP are multilayer perceptrons and each layer outputs $m$-dimensional features followed by a ReLU activation. For instance, the output of 2-MLP is $g(s_i) = \sigma(W_2\sigma(W_1 s_i))$,

Table 3: Top-1 accuracy (%) on CIFAR-100 using deep projectors. 2x2-MLP and 2x3-MLP indicate two-layer MLPs with wider hidden layers.

| Teacher | VGG13 | ResNet32x4 |
|---|---|---|
| Student | VGG8 | ResNet8x4 |
| 2-MLP | 73.31 | 75.12 |
| 2x2-MLP | 73.31 | 75.00 |
| 2x3-MLP | **73.37** | **75.13** |

where $W_1 \in \mathbb{R}^{m \times d}$ and $W_2 \in \mathbb{R}^{m \times m}$ are weighting matrices. It is shown that simply increasing the depth of the projector does not improve the performance of the student and tends to degrade the effectiveness of the projector. We hypothesize that with the increase of depth, the teacher's features can be overfitted by the projector. Besides, we also evaluate the effect of deep projectors with wider hidden layers as shown in Table 3. In this experiment, the hidden dimensions of 2x2-MLP and 2x3-MLP are $2 \times m$ and $3 \times m$, respectively. Experimental results demonstrate that the improvement is marginal by increasing the width of deep projectors.

Table 4: Diversity of projectors in the proposed distillation framework on CIFAR-100 with teacher-student pair ResNet32x4-ResNet8x4.

| Epoch | 10 | 40 | 80 | 120 | 160 | 200 | 240 |
|---|---|---|---|---|---|---|---|
| L2 distance | 440.93 | 394.57 | 372.82 | 360.69 | 298.86 | 213.29 | 206.32 |

Table 5: Top-1 classification accuracy and standard deviation (%) on CIFAR-100 with different teacher-student pairs.

| Teacher | WRN-40-2 | VGG13 | ResNet32x4 | ResNet50 | ResNet50 |
|---|---|---|---|---|---|
| Student | WRN-16-2 | VGG8 | ResNet8x4 | VGG8 | MobileNetV2 |
| Teacher | 75.61 | 74.64 | 79.42 | 79.34 | 79.34 |
| Student | 73.22±0.13 | 70.74±0.31 | 72.93±0.28 | 70.74±0.31 | 65.03±0.09 |
| KD | 74.92±0.28 | 72.98±0.19 | 73.33±0.25 | 73.81±0.13 | 67.35±0.32 |
| CC | 73.56±0.26 | 70.71±0.24 | 72.97±0.17 | 70.25±0.12 | 65.43±0.15 |
| SP | 73.83±0.12 | 72.68±0.19 | 72.94±0.23 | 73.34±0.34 | 68.08±0.38 |
| RKD | 73.35±0.09 | 71.48±0.05 | 71.90±0.11 | 71.50±0.07 | 64.43±0.42 |
| FitNets | 73.58±0.32 | 71.02±0.31 | 73.50±0.28 | 70.69±0.22 | 63.16±0.47 |
| FT | 73.25±0.20 | 70.58±0.08 | 72.86±0.12 | 70.29±0.19 | 60.99±0.37 |
| CRD | 75.48±0.09 | 73.94±0.22 | 75.51±0.18 | 74.30±0.14 | 69.11±0.28 |
| SRRL | 75.59±0.17 | 73.44±0.07 | 75.33±0.04 | 74.23±0.08 | 68.41±0.54 |
| Ours | **76.02±0.10** | **74.35±0.12** | **76.08±0.33** | **74.58±0.22** | **69.81±0.42** |

## 4.2 Diversity of Projectors

This section investigates the diversity of projectors in the proposed ensemble-based distillation framework. In this experiment, we set $q = 2$ for demonstration. The L2 distances between two projectors at different epochs are given in Table 4. We can see that the diversity of projectors gradually decreases with the increase of training epochs and appears to converge after 200 epochs. In Supplementary Material, we discuss how to promote the diversity of projectors from the perspectives of using different initialization methods and adding a regularization term.

## 4.3 Results on CIFAR-100

Table 5 reports the experimental results on CIFAR-100 with five teacher-student pairs. We run our method for three times with different seeds and obtain the average accuracy. Since CID requires different hyper-parameters for different pairs to achieve good performance, we omit it for comparisons on CIFAR-100. Among the projector-free distillation methods, the logit-based method KD shows

Table 6: Top-1 accuracy (%) of our method with different activation functions on CIFAR-100.

| Teacher | VGG13 | ResNet32x4 |
|---|---|---|
| Student | VGG8 | ResNet8x4 |
| w/ ReLU (Ours) | 74.35±0.12 | 76.08±0.33 |
| w/ GELU | **74.39±0.18** | **76.32±0.27** |
| w/o activation | 73.46±0.42 | 75.04±0.37 |

better performance compared to the similarity-based methods CC, SP and RKD. Furthermore, KD outperforms the projector-based methods FitNets and FT in most cases. Since FitNets is designed to distill the intermediate features, its performance is unstable for teacher-student pairs using different architectures. FT uses an auto-encoder as the projector to extract latent representations of the teacher, which may disturb the discriminative information to some extent and consequently degrade the performance. The recently proposed feature distillation methods CRD and SRRL show competitive performance compared to the previous methods by distilling the last layer of features. By harnessing the power of both distilling the last features and projector ensemble, the proposed method consistently achieves the highest accuracy on CIFAR-100.

We further investigate the effect of the activation function. Table 6 shows that the addition of the ReLU activation function has a significant positive impact on the performance of the proposed method. The reason is that the lack of an activation function and the non-linearity introduced by it limits the diversity of the projectors, as a group of linear projections can be mathematically reduced to a single linear projection through sum-pooling, which will degrade the distillation performance. Recently, Gaussian Error Linear Units (GELU) [10] has received attention because of its effectiveness on

Table 7: Classification accuracy (%) on ImageNet with different teacher-student pairs (a) ResNet34-ResNet18, (b) ResNet50-MobileNet, (c) DenseNet201-ResNet18 and (d) DenseNet201-MobileNet.

| Pair | Accuracy | Student | KD | SP | CRD | SRRL | CID | Ours | Teacher |
|------|----------|---------|-----|-----|-----|------|-----|------|---------|
| (a)  | Top-1    | 69.75   | 70.83 | 70.94 | 70.85 | 71.71 | 71.86 | **71.94** | 73.31 |
|      | Top-5    | 89.07   | 90.15 | 89.83 | 90.12 | 90.58 | 90.63 | **90.68** | 91.41 |
| (b)  | Top-1    | 69.06   | 70.65 | 70.14 | 71.03 | 72.58 | 72.25 | **73.16** | 76.13 |
|      | Top-5    | 88.84   | 90.26 | 89.64 | 90.16 | 91.05 | 90.98 | **91.24** | 92.86 |
| (c)  | Top-1    | 69.75   | 70.38 | 70.75 | 70.87 | 71.76 | 71.99 | **72.29** | 76.89 |
|      | Top-5    | 89.07   | 90.12 | 90.01 | 89.86 | 90.80 | 90.64 | **90.99** | 93.37 |
| (d)  | Top-1    | 69.06   | 69.98 | 70.34 | 70.82 | 72.28 | 71.90 | **73.24** | 76.89 |
|      | Top-5    | 88.84   | 89.93 | 89.63 | 90.09 | 90.90 | 90.97 | **91.47** | 93.37 |

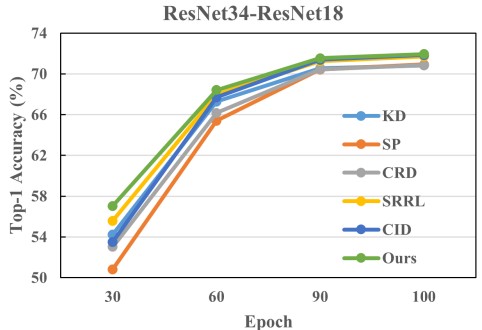 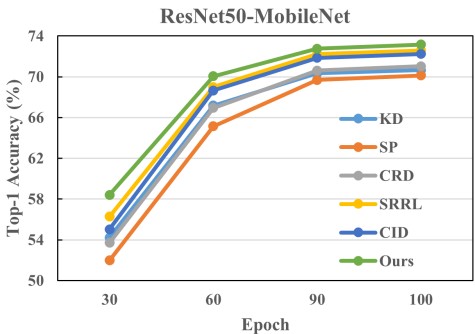

Figure 3: Top-1 accuracy of different methods on ImageNet with different number of epochs and different teacher-student pairs.

Transformer [5, 6]. We replace ReLU with GELU in the proposed method to test the performance. It is shown that the performance of our method can be further improved by using GELU as activation.

## 4.4 Results on ImageNet

The performance of the students distilled by different methods are listed in Table 7. Compared to the settings in previous methods [30, 33, 4], we introduce more teacher-student pairs for evaluation in this experiment so that the generalizability of different methods can be better evaluated. As presented in the table, feature distillation methods (CRD, SRRL, CID and our method) outperform both the logit-based method (KD) and the similarity-based method (SP) in most cases.

One major difference between CRD and the other feature distillation methods is the way of feature transformation. CRD transforms teacher and student features simultaneously while the other methods only transform student features. By solely mapping student features into the teacher's space, the original teacher feature distribution can be preserved without losing discriminative information. Therefore, SRRL, CID and our method obtain better performance than CRD. Besides, our method consistently outperforms the state-of-the-art methods SRRL and CID with different teacher-student pairs. With pair DenseNet201-MobileNet, the proposed method obtains 0.96% and 0.50% improvements compared to the second best method in terms of top-1 and top-5 accuracy, respectively. MobileNet (4.2M parameters) distilled by our method can obtain similar performance and reduce about 80% of the parameters compared to the ResNet34 (21.8M parameters). Figure 3 reports the top-1 accuracy of different methods with different training epochs. It is shown that the proposed method converges faster than the other distillation methods.

In [21], the authors observe that a better teacher may fail to distill a better student. Such phenomenon also exists in Table 7. For example, compared to the pair ResNet50-MobileNet, most of the methods distill a worse student by using a better network DenseNet201 as the teacher. One plausible explanation for this phenomenon is that the knowledge of a better teacher is more complex and is more difficult to learn. To alleviate this problem, TAKD [21] introduces some smaller assistant networks to facilitate training. Densely Guided Knowledge Distillation (DGKD) [28] further extends TAKD

Table 9: Training times (in second) of one epoch and peak GPU memory usages (MB) of different methods on ImageNet with teacher-student pair DenseNet201-ResNet18.

| Method | KD | SP | CRD | SRRL | CID | Ours |
|---|---|---|---|---|---|---|
| Time | 2,969 | 2,989 | 3,158 | 3,026 | 3,587 | 2,995 |
| Memory | 11,509 | 11,509 | 15,687 | 11,991 | 12,021 | 11,523 |

with dense connections between different assistants. However, the training costs of these methods are greatly increased by using the assistant networks. As shown in Table 7, the proposed method has the potential to alleviate this problem without introducing the additional networks.

We compare the training costs of different methods in Table 9. Since KD and SP are projector-free methods, their training costs are lower than that of the feature distillation methods. The training cost of our method is slightly higher than KD and SP because we use multiple projectors to improve the optimization of the student. On the other hand, the proposed method only uses a naive direction alignment loss to distill the knowledge. Therefore, the computation complexity and memory usages are lower compared to the other feature-based methods.

Two recently proposed methods, namely Attention-based Feature Distillation (AFD) [15] and Knowledge Review (KR) [2] are also introduced for comparisons as reported in Table 8. Unlike methods that utilize the last layer of features for distillation (CRD, SRRL, CID and ours), AFD and KR propose to extract information from multiple layers of features. Table 8 shows that the proposed method performs better than AFD and KR with different teacher-student pairs in terms of the comparisons of top-1 and top-5 accuracy, which in-

Table 8: Comparisons of the proposed method and distillation methods using multiple layers of features.

| Teacher | ResNet34 | | ResNet50 | |
|---|---|---|---|---|
| Student | ResNet18 | | MobileNet | |
| Acc | Top-1 | Top-5 | Top-1 | Top-5 |
| Teacher | 73.31 | 91.41 | 76.13 | 92.86 |
| Student | 69.75 | 89.07 | 69.06 | 88.84 |
| AFD | 71.38 | – | – | – |
| KR | 71.61 | 90.51 | 72.56 | 91.00 |
| Ours | **71.94** | **90.68** | **73.16** | **91.24** |

dicates that using the last layer of features is sufficient to obtain good distillation performance on ImageNet.

## 5 Conclusion

This paper studies the effect of the projector in feature distillation and proposes a projector ensemble-based architecture to improve the feature projection process from the student to the teacher. We first investigate the phenomenon that the addition of a projector improves the distillation performance even when the feature dimensions of the student and the teacher are the same. From the perspective of multi-task learning, we speculate that the student network without a projector tends to overfit the teacher's feature space and pays less attention to the task of discriminative feature learning for classification. By imposing a projector on the student network to mitigate the overfitting problem, the student's performance can be increased. Based on the positive effect of the projector in feature distillation, we propose an ensemble of projectors for further improvement. Empirical results on ImageNet and CIFAR-100 show that our method consistently achieves competitive performance with different teacher-student combinations, compared to other state-of-the-art methods.

**Limitations and future work.** In addition to the image classification task, the proposed method can be further applied in other downstream tasks (e.g., object detection and semantic segmentation), which can be explored in future work. Besides, the proposed method focuses on using the direction alignment loss for distillation. How to effectively and efficiently integrate logits and similarity information into the proposed framework is a potential research direction.

## 6 Acknowledgments

This work is partially supported by Australian Research Council DE200101610 and CSIRO's Research Plus Science Leader Project R-91559.

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
