# Supplementary Material for "Improved Feature Distillation via Projector Ensemble"

## 1 Ablation Studies

In this section, we further investigate the effectiveness of the proposed method when the feature dimensions of the student and teacher are different. In this experiment, we use teacher-student pair ResNet50-MobileNet (the teacher outputs 2048-dimensional feature vectors and the student outputs 1024-dimensional feature vectors) and report the top-1 classification accuracy (%) in Table 1. Experimental results show that the proposed projector ensemble method can consistently improve the distillation performance when the feature dimensions of the student and teacher are different.

Table 1: Top-1 classification accuracy (%) on ImageNet. 1-Proj, 2-Proj, 3-Proj and 4-Proj indicate the number of single-layer projectors in the ensemble.

|  | 1-Proj | 2-Proj | 3-Proj | 4-Proj |
|---|---|---|---|---|
| Accuracy | 72.75 | 73.15(+0.4) | 73.16(+0.41) | **73.29(+0.54)** |

## 2 Initialization of Projectors

In our experiments, we find that simply initializing different projectors with different seeds and the default initialization method of linear layer in Pytorch is sufficient to yield good performance. Therefore, we stick to this strategy to make the proposed method as simple as possible. We also compare the distillation performance by using different initialization methods in Table 2. Experimental results show that mixing different initialization methods has a slight impact on the performance and is a potential way to further improve the distillation performance. In this experiment, we use ResNet32x4-ResNet8x4 on CIFAR-100 and report the top-1 classification accuracy.

Table 2: Top-1 classification accuracy (%) on CIFAR-100.

|  | Kaiming Ini. [1] | Orthogonal Ini. | Ours(Default Ini.) | Mixing Different Ini. |
|---|---|---|---|---|
| Accuracy | 75.78 | 76.12 | 76.08 | **76.27** |

## 3 Diversity of Projectors

To promote the diversity of projectors, we add a regularization term to explicitly maximize the L2 distances between projectors as follows:

$$\mathcal{L}_{total} = \mathcal{L}_{CE} + \alpha\mathcal{L}_{MDA} - \beta\mathcal{L}_{PD}, \tag{1}$$

where $\beta$ is a hyper-parameter and $\mathcal{L}_{PD} = ||W_1 - W_2||^2$. In this experiment, we set $q = 2$ for simplicity and use ResNet32x4-ResNet8x4 on CIFAR-100 for demonstration. The L2 distances between projectors with and without the regularization term is shown in Table 3. From some preliminary results, adding the regularization term can marginally increase the projectors' diversity after 160 epochs and consequently improve the performance of the student.

36th Conference on Neural Information Processing Systems (NeurIPS 2022).

Table 3: L2 distances between projectors and classification accuracy (%) on CIFAR-100.

|  | 40 Epochs | 80 Epochs | 160 Epochs | 240 Epochs | Top-1 Accuracy |
|---|---|---|---|---|---|
| w/o $\mathcal{L}_{PD}$ | 394.57 | 372.82 | 298.86 | 206.32 | 75.66 |
| w/ $\mathcal{L}_{PD}$ | 384.53 | 368.17 | 301.39 | 211.61 | **75.86** |

# 4  Transferability of Students

We transfer the knowledge of MobileNet distilled by ResNet50 on ImageNet to CUB200 [3] and Cars196 [2] datasets. We freeze the parameters of networks and re-train the last linear classifiers. The generalization performance of networks distilled by different methods is shown in Table 4. In this experiment, we report the top-5 classification accuracy (%) of different methods. Experimental results indicate that the proposed distillation method can significantly improve the generalization ability of networks on downstream tasks compared to the SOTA methods.

Table 4: Top-5 classification accuracy (%) on different datasets.

|  | w/o distillation | CRD | CID | SRRL | Ours |
|---|---|---|---|---|---|
| CUB200 | 89.62 | 89.61 | 90.21 | 90.26 | **91.14** |
| Cars196 | 79.33 | 79.62 | 77.59 | 80.41 | **82.27** |

# 5  Training Costs

Table 5 reports the training times and memory usages of the proposed method with different number of projectors. We can see that the training times and memory usages of our method will slightly increase with the increase of the number of projectors. On the other hand, in our experiments, the proposed method uses three projectors and outperforms the SOTA methods. With three projectors, the training times and memory usages of our method are relatively lower than existing feature distillation methods (e.g. CRD, SRRL and CID).

Table 5: Training times (in second) of one epoch and peak GPU memory usages (MB) on ImageNet with teacher-student pair DenseNet201-ResNet18.

|  | CRD | SRRL | CID | 1-Proj | 2-Proj | 3-Proj | 4-Proj |
|---|---|---|---|---|---|---|---|
| Time | 3,158 | 3,026 | 3,587 | 2,995 | 2,988 | 2,995 | 3,004 |
| Memory | 15,687 | 11,991 | 12,021 | 11,475 | 11,523 | 11,523 | 12,215 |