# OpenReview forum: "Improved Feature Distillation via Projector Ensemble"
_NeurIPS.cc/2022/Conference — NeurIPS 2022 Accept_

### Official Review · Reviewer_EwZs · 2022-06-30

**Rating:** 5
**Confidence:** 5
**Soundness:** 2 fair
**Presentation:** 3 good
**Contribution:** 2 fair

**Summary:**

This work proposed a new feature distillation method via projector ensemble. In particular, different from the traditional single-projector method, it introduces multiple projectors and computes the average output to perform KD. Further, two different designs including a single-layer projector and MLP projector are discussed and compared. Experiments on popular datasets are conducted to verify the effectiveness of the method.

**Questions:**

Please refers to the weaknesses mentioned above.

**Limitations:**

As mentioned in the paper, the proposed method is only verified on classification. Its effectiveness on other tasks can be further explored.

**Strengths And Weaknesses:**

The strengths are listed below:
(1)	The proposed method achieves significant performance improvement.
(2)	This proposed method is simple and easy to be reproduced.

The weaknesses are listed below:
(1)	The novelty is limited. The single-layer projector is widely used in KD. Further, a simple extension from single-layer to multi-layer is rather simple. It is easy to come up with this idea.
(2)	Lack of references in all of the tables. It is recommended to add references of comparing methods in tables.

---

> ### Author Response · Authors · 2022-08-02
> **Response to Reviewer EwZs**
>
> The authors are thankful for the reviewer's comments.
>
> **Q1: The novelty is limited. The single-layer projector is widely used in KD. Further, a simple extension from single-layer to multi-layer is rather simple. It is easy to come up with this idea.**
> A1: We respectfully disagree with the reviewer's comments. We have to emphasize that this paper **DOES NOT** propose to extend the projector from single-layer to multi-layer. In fact, as shown in Fig.4 and we mentioned in Line 204 in our manuscript, simply increasing the number of layers of the projector tends to degrade the distillation performance. Instead, this paper proposes to ensemble a series of single-layer projectors. The proposed idea is simple but efficient and effective . Besides, to the best of our knowledge, this is the first paper using the projector ensemble strategy to improve the feature distillation performance. There are multiple factors may prevent researchers from using the projector ensemble strategy (Please refer to A1 to Reviewer 52Wo ).
>
> **Q2: Lack of references in all of the tables. It is recommended to add references of comparing methods in tables.**
> A2: Thanks for the reviewer's suggestion.

---

### Official Review · Reviewer_2exr · 2022-07-11

**Rating:** 6
**Confidence:** 5
**Soundness:** 4 excellent
**Presentation:** 3 good
**Contribution:** 3 good

**Summary:**

The paper presents a feature matching-based distillation method that makes use of a set of feature projectors to better align the features of the student and teacher network.
The authors show that such an ensemble of projectors can improve the distillation performance further even if the student and teacher have the same feature dimension.
Some analysis is conducted to show the reason of introducing a projector can benefit the student network learning.
Experiments were performed on benchmark datasets using various teacher-student network structure pairs to compare with different methods.

**Questions:**

1. Line 251, which table is this referring to? Is it table 1?
2. Is there any limitation in the training of an ensemble of projectors? Is the training time impacted by the number of projectors and/or the depth of the projectors?

**Limitations:**

The authors have a section for discussing limitations and future works. However, it did not directly address the limitation of the multiple projectors, e.g. training time, memory usage, etc.

**Strengths And Weaknesses:**

Strengths:
- The paper is well written with comprehensive experiments.
Weakness:
- The idea is simple but the authors should explore more options for an ensemble of projectors, e.g. ensemble of various structure projectors.

---

> ### Author Response · Authors · 2022-08-02
> **Response to Reviewer 2exr**
>
> We thank reviewer for the positive comments.
>
> **Q1: The idea is simple but the authors should explore more options for an ensemble of projectors, e.g. ensemble of various structure projectors.**
> A1: In our experiments, we find that integrating projectors with similar architecture yields better performance. We select two commonly-used projector's architecture for demonstration (i.e., 1x1 convolutional kernel and our single-layer MLP). We report the top-1 classification accuracy on CIFAR-100 with pair ResNet32x4-ResNet8x4 in the following table. Besides, we also investigate the performance of our method by using different initialization methods (Please refer to A5 to Reveiewr 52Wo).
>
> w/o Projector |Conv |MLP |Conv+Conv |MLP+MLP |Conv+MLP
> -|-|-|-|-|-
> 73.66  |73.87  |75.14   |74.73  |**76.08** |75.96
>
>
> **Q2: Is there any limitation in the training of an ensemble of projectors? Is the training time/memory usage impacted by the number of projectors?**
> A2: The reviewer raises an interesting question. The following table (similar to Table 3 in our manuscript) evaluates the training costs and memory usages of different distillation methods and the proposed method with different number of projectors. We can see that the training cost and memory usage of our method will slightly increase with the increase of the number of projectors. On the other hand, in our experiments, the proposed method uses three projectors and outperforms the SOTA methods. With three projectors, the training cost and memory usage of our method are relatively lower than existing methods as shown in the following table.
>
> Complexity|CRD |SRRL |CID |1-Proj |Ours(2-Proj)|Ours(3-Proj) |Ours(4-Proj)
> -|-|-|-|-|-|-|-
> Times(s)  |3,158 |3,026 |3,587 |2,995 |2,988 |2,995 |3,004
> Memory(MB) |15,687  | 11,991 |12,021  |11,475 |11,523 |11,523 |12,215
>
> **Q3: Line 251, which table is this referring to? Is it table 1?**
> A3: Yes, it is table 1. We will clarify this in the revised manuscript.

---

> > ### Comment · Reviewer_2exr · 2022-08-09
> > **Final rating remain unchanged**
> >
> > The authors provided more explanation and experiments in their response to reviewers. This is very helpful for reviewers to understand and evaluate this work. I would suggest the authors try to include those additional materials in the paper or supplementary materials so that the reader can have better understanding.
> >
> > Another clarification for response of Q2, how are the training cost and memory usages computed? Is it the time to train 1 epoch? Is it the peak GPU memory with batch size of how many samples?

---

> > > ### Author Response · Authors · 2022-08-10
> > > **Response to Reviewer 2exr**
> > >
> > > We sincerely appreciate your support. In the table for Q2, we use teacher-student pair DenseNet201-ResNet18 on ImageNet for demonstration. In this table, we report the training times of one epoch of different methods and record the peak GPU (an NVIDIA V100 GPU) memory usages of different methods with batch size of 256. Following the reviewer's suggestion, we will clarify these details in the revised manuscript and include the additional experiments in the supplementary materials.

---

### Official Review · Reviewer_Y25R · 2022-07-12

**Rating:** 5
**Confidence:** 4
**Soundness:** 2 fair
**Presentation:** 2 fair
**Contribution:** 2 fair

**Summary:**

This is an experiment-driven paper. The authors do lots of experiments and comparisons to demonstrate the effectiveness of feature projectors in feature distillation.

**Questions:**

See weaknesses.

**Limitations:**

See weaknesses.

**Strengths And Weaknesses:**

Strengths:

The authors do lots of experiments and comparisons to show the superiority of feature projectors in feature distillation.

Weaknesses:

1. This is an experiment-driven paper. The authors do lots of experiments and give empirical analyses. However, I still think the contributions are not enough for NeurIPS.
2. The authors should re-organize the Method section. It is not obvious from one equation to the next. And the notations are somewhat confused.
    - The notations $s^p$ and $W^p$ is confused. I suggest to use $t$ as the $t$-th iterations and use $W_p$ to represent the parameters of projector. It is more clear.
    - The Eq. (2), Eq. (3) and Eq. (4) should be more clear. It is not easy to understand, especially for the junior researchers. For example, Eq. (3) can be denoted as $\frac{\partial L_{DA}}{\partial s_i^p}=\frac{\partial L_{DA}}{\partial g(s_i^p)}\frac{\partial g(s_i^p)}{\partial s_i^p}$. The last term of Eq. (4) can be denoted as $\frac{\partial L_{DA}}{\partial g(s_i^p)}\frac{\partial g(s_i^p)}{\partial W^p}$.
3. The hypothesis (Line 125-126) is a little coarse. I think the main contribution in this paper is the analysis about the effectiveness of the projector. Therefore, the authors should give deeper analyses.
4. In experiments, table is more suitable and clear to demonstrate the performance than figure. I recommend the authors use tables instead of Fig. 3 and Fig. 4.
5. Considering the feature distillation, some experiments about CUB-200-2011, Cars-196 and SOP should be included.

---

> ### Author Response · Authors · 2022-08-02
> **Response to Reviewer Y25R**
>
> The authors are thankful for the reviewer's comments.
>
> **Q1: This is an experiment-driven paper. The authors do lots of experiments and give empirical analyses. However, I still think the contributions are not enough for NeurIPS.**
> A1: The authors would like to argue that most of existing distillation methods are experiment-driven since it is difficult to give deeper insight to the mechanism of knowledge distillation as well as deep learning. For contributions: (1) Technically and conceptually, differ from existing feature distillation methods that focus on the design of loss functions and the selection of the distilled layer, this paper demonstrates that it is also feasible to achieve promising distillation performance by simply modifying the architecture of the projector, which presents a new perspective for the improvement of feature distillation; (2) Experimentally, we conduct comprehensive evaluation to illustrate the effectiveness of the proposed projector ensemble and show that the proposed method outperforms SOTA feature distillation methods with less training complexity. Based on the simplicity and effectiveness of the proposed method, we believe that this paper will be beneficial to the related research.
>
> **Q2: The authors should re-organize the Method section. It is not obvious from one equation to the next. And the notations are somewhat confused.**
> A2: Thanks for the reviewer's suggestion! We will improve the readability of the manuscript.
>
> **Q3: The hypothesis (Line 125-126) is a little coarse. I think the main contribution in this paper is the analysis about the effectiveness of the projector. Therefore, the authors should give deeper analysis.**
> A3: We give more analysis about the diversity and behaviors of projectors. Please refer to A2 and A3 to Reviewer n8Zt.
>
> **Q4: In experiments, table is more suitable and clear to demonstrate the performance than figure. I recommend the authors use tables instead of Fig. 3 and Fig. 4.**
> A4: We thank the reviewer for this advice. We will take this recommendation.
>
> **Q5: Considering the feature distillation, some experiments about CUB-200-2011, Cars-196 and SOP should be included.**
> A5: Thanks for providing related datasets. Due to the time limitation, we only test the performance of different methods on CUB200 and Cars196 datasets. We transfer the knowledge of MobileNet trained on ImageNet to CUB200 and Cars196 datasets. We freeze the parameters of networks and re-train the last linear classifiers. The generalization performance of networks distilled by different methods is shown in the following table. In this experiment, we report the top-5 classification accuracy (%) of different methods. Experimental results indicate that the proposed distillation method can significantly improve the generalization ability of networks on downstream tasks compared to the SOTA methods.
>
> Datasets |w/o distillation |CRD |CID |SRRL |Ours
> -|-|-|-|-|-
> CUB200  |89.62 |89.61 |90.21  |90.26  |**91.14**
> Cars196 |79.33 |79.62 |77.59  |80.41  |**82.27**

---

> > ### Comment · Reviewer_Y25R · 2022-08-08
> > **Final rating**
> >
> > After reading the rebuttal and other reviews, I decide to improve the final rating of the submission.

---

### Official Review · Reviewer_n8Zt · 2022-07-21

**Rating:** 5
**Confidence:** 4
**Soundness:** 3 good
**Presentation:** 3 good
**Contribution:** 2 fair

**Summary:**

This paper proposes a feature distillation method based on an ensemble of multiple projectors.  The model uses multiple projectors to project the features of a student model, and the Direction Alignment loss are calculated between the multiple projected features and the teacher features. Some analysis are given to explain why projector is helpful. Experimental results show that the proposed method outperforms previous competitors.

**Questions:**

1. I suggest the authors give some visualization results or experimental analysis of the projector behaviors during and after training. If different projectors show significant differences, it would be better to give some theoretical analysis about what brings them variety.
2. The theoretical analysis in section 3.1 needs to be clearer. Also, the authors need to give explanations of why more projectors are better under their feature gradients perspective.

**Limitations:**

The authors have addressed the limitations and potential negative societal impacts.

**Strengths And Weaknesses:**

Strengths:

1. This paper focuses on the projectors in feature distillation models and proposes a simple but effective ensemble method. This angle is interesting and the proposed model can be widely applicated.

2. The writing is clear and easy to understand.

3. Experimental results show that the proposed model obtains promising results.

Weaknesses:

1. The ensemble strategy is too simple. The authors simply add all the projected features together.  In fact, ensemble strategy plays a critical role in ensemble learning. In many situations, simply averaging the outputs of weak learners cannot bring improvements.

2. This paper lacks discussions about the projector behaviors, especially their diversity. The authors only give a simple description of "projectors with different initialization would provide different transformed features, which is beneficial to the generalizability of the student".  However, different initialization cannot guarantee enough variety between projectors. Moreover, the different projected features are simply added together to calculate losses. There is no special architecture and training designs to promote the projector diversity. In experiments, there is also no experimental analysis of the projector behaviors.

3. The theoretical analysis is not clear enough. In section 3.1, the authors try to analyze why projector is helpful by comparing the feature gradients with and without projector. The explanation is that the non-linear transformation updated from previous data helps to better capture the global feature distribution. This is not convincing enough to me. Furthermore, this explanation cannot explain why multiple projectors are better.

---

> ### Author Response · Authors · 2022-08-02
> **Response to Reviewer n8Zt**
>
> The authors are thankful for the reviewer's comments.
>
> **Q1: The ensemble strategy is too simple. The authors simply add all the projected features together. In fact, ensemble strategy plays a critical role in ensemble learning. In many situations, simply averaging the outputs of weak learners cannot bring improvements.**
> A1: We would like to argue that being simple should not be viewed as a weakness of the proposed method. Instead, we believe that simplicity is a strength of our method from the perspective of Occam's Razor. According to the experimental results in the submitted manuscript, the ensemble strategy can effectively improve the feature distillation performance and the proposed distillation framework can consistently outperform SOTA methods.
>
> **Q2: This paper lacks discussions about the projector behaviors, especially their diversity. Different initialization cannot guarantee enough variety between projectors. Moreover, the different projected features are simply added together to calculate losses. There is no special architecture and training designs to promote the projector diversity. In experiments, there is also no experimental analysis of the projector behaviors.**
> A2: We thank the reviewer for this insightful comment. We investigate the diversity and behaviors of projectors from the following two perspectives: (1) In the following table, we compute the L2 distance between two different projectors in our ensemble distillation framework. We can see that the diversity of projectors gradually decreases with the increase of training epochs. However, projectors are still different from each other during training even if we do not use special training designs;
> Epoch |40 Epochs | 80 Epochs | 160 Epochs |240 Epochs
> -|-|-|-|-
> L2 distance |394.57|372.82 |298.86 |206.32
>
> (2) We compute the average cosine similarities between teacher and student features transformed by two different projectors. We report the the top-3 categories with the largest average cosine similarities obtained by different projectors in the following table. From this table, we can see that different projectors prone to fit different teacher features of different classes (e.g., Proj-1 tends to fit samples of the 4-th class, Proj-2 tends to fit samples of the 22-th class), which indicates the diversity of projectors during and after training.
>
> Epoch |40 Epochs | 80 Epochs | 160 Epochs |240 Epochs
> -|-|-|-|-
> Proj-1 |(26,93,4) |(3,4,11)|(4,55,26) |(4,55,26)
> Proj-2 |(22,45,44) |(45,98,35) |(45,22,72) |(22,45,44)
>
>
> **Q3: The theoretical analysis in section 3.1 needs to be clearer. Also, the authors need to give explanations of why more projectors are better under their feature gradients perspective.**
> A3: In section 3.1, we hypothesize that the student network may better capture the global feature distribution by introducing a projector to preserve data information during back propagation. However, since projectors with random initialization may contain bias as discussed in A2, we propose to integrating multiple projectors. By taking the average of different projectors, the distribution of the projected features will be smoother. We empirically verify this hypothesis by computing the standard deviation (the lower the better) of average cosine similarities between teacher and student features in the following table. From this table, we can see that by imposing a projector to assist distillation, the student network can better capture global data distribution of the teacher according to its lower standard deviation compared to the student w/o Proj. Furthermore, by using an ensemble of projectors, the performance can be further improved.
>
> w/o Proj. |One Proj. |Two Proj. Ensemble |Three Proj. Ensemble
> -|-|-|-
> 0.024 |0.018 |0.017 |0.016

---

> > ### Comment · Reviewer_n8Zt · 2022-08-07
> > **Rating is improved to 5 (borderline accept)**
> >
> > The authors' comments have addressed most of my concerns.
> >
> > I have one more question regarding Q2. In the above comments, it is shown that the distances between projectors decrease during training. I wonder the corresponding performance changes. In other words, if we continue the training process even the model performance reaches saturation (prone to overfit), what will happen to the distance between projectors? I suggest the authors add some analysis about this.

---

> > > ### Author Response · Authors · 2022-08-09
> > > **Response to Reviewer n8Zt**
> > >
> > > We sincerely appreciate your support. Regarding your penetrating question, we train the additional 120 epochs based on the original 240 epochs and report the corresponding L2 distances between two projectors w.r.t different epoch numbers (with a step of 20 epochs), in the following table. In this table, we observe that the parameters of projectors tend to converge after 300 epochs and the L2 distance between them basically remain unchanged from 300 epochs to 360 epochs.
> > >
> > > Epoch |  240 Epochs  |260 Epochs |280 Epochs |300 Epochs |320 Epochs |340 Epochs  |360 Epochs
> > > |  ----  | ----  |----  | ----  |----  | ----  |----  | ----  |
> > > L2 distance | 206.32  |205.15|203.98  |202.80  |202.68  |202.55 |202.43
> > >
> > > On the other hand, inspired by the reviewer's comments, we simply add a regularization term to explicitly promote the diversity between projectors. The goal of this regularization term is to maximize the L2 distance between projectors as a basic strategy. The results are shown in the following table. From some preliminary results, adding the regularization term can marginally increase the projectors' diversity after 160 epochs and improve distillation performance (approximately 0.2% top-1 accuracy on CIFAR100) . More sophisticated designs that can leverage the diversity of projectors are worth paying further research attention to, which will be included as a separate analysis in the revised manuscript. We thank the reviewer for providing this inspiring suggestion. Any further discussions, recommendations and support are appreciated very much.
> > >
> > > Epoch| 40 Epochs	| 80 Epochs	| 160 Epochs	| 240 Epochs
> > > |  ----  | ----  |----  | ----  |----  |
> > > w/o regularization |394.57	|372.82	|298.86	|206.32
> > > w/ regularization |384.53 |368.17 |301.39 |211.61

---

### Official Review · Reviewer_52Wo · 2022-07-21

**Rating:** 5
**Confidence:** 3
**Soundness:** 2 fair
**Presentation:** 3 good
**Contribution:** 3 good

**Summary:**

The authors discuss the phenomenon that using a projector on a student’s feature can improve the performance of distillation when the feature dimensions are the same.
They focus on the projector between teachers and students, which is unnoticed in the past.
Then the authors propose a simple ensemble method to further improve the performance.
They conduct comprehensive experiments in this paper and the authors claim the superiority of their proposal.
A complete story, from the phenomenon to the essence, which is simple and effective.


**Questions:**


“the student network benefits from a projector even if the feature dimensions of the student and teacher are the same” , Is this conclusion first discovered and proposed by the author of this paper? The word used in the abstract is “observe”.

Initialization of different projectors is not clear, do different initialization methods have a big impact on the experimental results?


**Limitations:**

As mentioned above in the Weakness.

**Strengths And Weaknesses:**

Strengths
1.	The method is simple and easy to implement.
2.	Experiments are detailed and code is provided.
3.	This paper is well written.
Weakness
1.	The novelty of the proposed method is limited. Are you sure that there are no more researchers who have used this method to improve the performance of their model?
2.	the number of projectors in the ensemble seems to be sensitive, how to determine this hyperparameter appropriately in practice?
3.	The ablation study does not provide the effect of different numbers of projectors on distillation when the feature dimensions are different.

---

> ### Author Response · Authors · 2022-08-02
> **Response to Reviewer 52Wo**
>
> We thank reviewer for the positive comments.
>
> **Q1: Are you sure that there are no more researchers who have used this method to improve the performance of their model?**
> A1: Yes. There are some factors that may prevent researchers from proposing the idea of projector ensemble. Firstly, as discussed in the manuscript, most of existing feature distillation methods pay more attention to the design of loss functions (e.g., CRD, CID and SRRL) and the selection of the distilled layer (e.g., AFD and KR). Therefore, the effect of projector is largely-ignored. Secondly, a more common way to modify the projector's architecture is to increase the number of layers. However, results in Fig.4 in our manuscript show that increasing the number of layers of the projector tends to degrade the distillation performance.
>
> **Q2: The number of projectors in the ensemble seems to be sensitive, how to determine this hyper-parameter appropriately in practice?**
> A2: Similar to the settings of hyper-parameters in previous methods, in our experiments, we determine the number of projectors via grid search and observe that the proposed method generally obtains good trade-off between distillation performance and training costs by using an ensemble of three projectors.
>
> **Q3: The ablation study does not provide the effect of different numbers of projectors on distillation when the feature dimensions are different.**
> A3: Experimental results in the following table show that the ensemble of projectors can consistently improve the distillation performance when the feature dimensions of the student and teacher are different. In this table, we use the teacher-student pair ResNet50-MobileNet (the teacher outputs 2048-dimensional features and the student outputs 1024-dimensional features) and report the top-1 classification accuracy (%) on ImageNet.
> |1-Proj |2-Proj |3-Proj |4-Proj|
> |:-: | :-: | :-:| :-:|
> |72.75 |73.15(+0.4) |73.16(+0.41) |73.29(+0.54) |
>
> **Q4: ''The student network benefits from a projector even if the feature dimensions of the student and teacher are the same'', is this conclusion first discovered and proposed by the author of this paper? The word used in the abstract is ''observe''.**
> A4: We believe that some of the related researchers may notice this phenomenon. However, to the best of our knowledge, this paper makes the first attempt to comprehensively study the effect of projectors in feature distillation.
>
> **Q5: Initialization of different projectors is not clear, do different initialization methods have a big impact on the experimental results?**
> A5: In our experiments, we find that simply initializing different projectors with different seeds and the default initialization method of linear layer in Pytorch is sufficient to yield good performance. Therefore, we stick to this strategy to make the proposed method as simple as possible. We also compare the distillation performance by using different initialization methods in the following table. Experimental results show that mixing different initialization methods has a slight impact on the performance and is a potential way to further improve the distillation performance. We thank the reviewer for providing this suggestion and will explore this in future work. The top-1 classification accuracy on CIFAR-100 with pair ResNet32x4-ResNet8x4 is as follows:
>
> Kaiming Ini. |Orthogonal Ini. |Ours(Default Ini.) |Mixing Different Ini.
> :-: | :-: | :-:| :-:
> 75.78  |76.12  |76.08  |76.27

---

### Meta-Review · Area_Chair_9Njj · 2022-08-25

**Recommendation:** Accept
**Confidence:** Certain

**Metareview:**

The paper received 5 positive reviews and the reviewers increased/remained their scores after the rebuttal. All the reviewers agree that the proposed method is simple yet effective, and the experiments are comprehensive.

Overall this work proposes an improved feature distillation method via projector ensemble. But I hope the authors will discuss the computational costs brought by multiple projectors clearly, as suggested by the reviewers.

**Award:**

No

---

### Decision · Program_Chairs · 2022-09-14

Accept